# Design, Synthesis and Mechanistic Studies of Novel Isatin-Pyrazole Hydrazone Conjugates as Selective and Potent Bacterial MetAP Inhibitors

**DOI:** 10.3390/antibiotics11081126

**Published:** 2022-08-19

**Authors:** Iram Irfan, Asghar Ali, Bharati Reddi, Mohd. Abrar Khan, Phool Hasan, Sarfraz Ahmed, Amad Uddin, Magdalena Piatek, Kevin Kavanagh, Qazi Mohd. Rizwanul Haque, Shailja Singh, Anthony Addlagatta, Mohammad Abid

**Affiliations:** 1Department of Biosciences, Jamia Millia Islamia, Jamia Nagar, New Delhi 110025, India; 2Division of Applied Biology, CSIR-Indian Institute of Chemical Technology, Hyderabad 500007, India; 3Academy of Scientific and Innovative Research (AcSIR), Ghaziabad 201002, India; 4Host-Parasite Interaction Biology Laboratory, Special Centre for Molecular Medicine, Jawaharlal Nehru University, New Delhi 110067, India; 5Department of Biology, Maynooth University, Maynooth, Co., W23F2H6 Kildare, Ireland

**Keywords:** Isatin-pyrazole hydrazone, ESKAPE, antibacterial, MDR, cytotoxicity, MetAP

## Abstract

Methionine aminopeptidases (MetAPs) are attractive drug targets due to their essential role in eukaryotes as well as prokaryotic cells. In this study, biochemical assays were performed on newly synthesized Isatin-pyrazole hydrazones (**PS1–14**) to identify potent and selective bacterial MetAP*s* inhibitors. Compound **PS9** inhibited prokaryotic MetAP*s*, i.e., *Mt*MetAP1c, *Ef*MetAP1a and *Sp*MetAP1a with ***K***_i_ values of 0.31, 6.93 and 0.37 µM, respectively. Interestingly, **PS9** inhibited the human analogue *Hs*MetAP1b with ***K***i (631.7 µM) about ten thousand-fold higher than the bacterial MetAPs. The in vitro screening against Gram-positive (*Enterococcus faecalis, Bacillus subtilis* and *Staphylococcus aureus*) and Gram-negative (*Pseudomonas aeruginosa, Klebsiella pneumonia* and *Escherichia coli*) bacterial strains also exhibited their antibacterial potential supported by minimum bactericidal concentration (MBC), disk diffusion assay, growth curve and time-kill curve experiments. Additionally, **PS6** and **PS9** had synergistic effects when combined with ampicillin (AMP) and ciprofloxacin (CIP) against selective bacterial strains. **PS9** showed no significant cytotoxic effect on human RBCs, HEK293 cells and *Galleria mellonella* larvae in vivo. **PS9** inhibited the growth of multidrug-resistant environmental isolates as it showed the MIC lower than the standard drugs used against selective bacterial strains. Overall, the study suggested **PS9** could be a useful candidate for the development of antibacterial alternatives.

## 1. Introduction

Antibiotic resistance is a global challenge [1,2,3]. Resistance to antimicrobials occurs naturally. In addition, factors such as overuse, misuse, substandard pharmaceuticals and inadequate use of antibiotics in veterinary, aquaculture, hospitals and agriculture could contribute for resistance [4,5,6,7,8]. The Centre for Disease Control and Prevention (CDC) reported at least 2.8 million individuals in the United State alone are infected with antibiotic-resistant bacteria or fungi each year, with over 35,000 deaths annually [9]. Antibiotic resistance is estimated to cost the European Union’s healthcare system more than €1.5 billion. In south Asia, India has highest infectious disease. The rates of antibiotic resistance are also alarming [10]. Recently it has been shown that more than 50% of the *Klebsiella* spp isolates from hospital-acquired infections in India are resistant to carbapenems and polymyxins resulting in about 70% deaths [11]. In comparison to non-resistant microorganisms, resistant bacteria cause double the number of deaths worldwide [12]. Many of these pathogens belong to the ESKAPE (*Enterococcus faecium, Staphylococcus aureus, Klebsiella pneumoniae, Acinetobacter buamannii, Pseudomonas aeruginosa* and *Enterobacter species*) group of priority pathogens (as defined by the World Health Organization) for which antibiotics are desperately needed [13,14,15]. These bacteria are a common cause of life-threatening nosocomial infections in critically ill and immunocompromised patients, and they possess drug resistance mechanisms [16]. As a result, the limitation of innovative and effective antimicrobials to control microbial growth and resistance has compelled us to create and evaluate new antimicrobial agents [17,18,19].

Development of small molecules by choosing molecular target is essential due to their safety and selectivity. Removal of amino terminal initiator methionine is an essential process in all living cells [20,21]. Methionine aminopeptidases (MetAPs) are the enzymes that co-translationally remove this residue. MetAP are dinuclear metalloproteases that hydrolyze the substrate to remove methionine from nascent proteins at *N*-terminus. Almost all microbes contain a single enzyme of this class. Furthermore, the importance of MetAPs has been revealed in several organisms including *Escherichia coli*, *Salmonella typhimurium* and *Mycobacterium tuberculosis*, whereby knockout of the MetAP gene lead to lethality or decreased viability [22,23,24]. Hence, microbial MetAP is recommended as one of the promising antibacterial drug targets.

To develop MetAPs inhibitors as antibacterial agents researchers are continuing their efforts to identify and develop new small semisynthetic/synthetic inhibitors to aid in the current global health care crisis [25]. To achieve this goal they have paid much attention to heterocycles containing a nitrogen atom, particularly pyrazoles and their derivatives [26]. Pyrazoles are an important class of compounds for new drug development that have attracted much attention. Like pyrazole, Isatin is also one of the most commonly encountered heterocyclic cores in medicinal chemistry due to its attractive pharmacological effects. It is an alkolide present in *Couroupita guianesis* trees [27]. A (*N*1-cyclopentyl-N2-(thiazol-2-yl)oxalamide) was shown to be potent *Ec*MetAP1 inhibitor with an IC_50_ value 67 nM with Co(II) as the cofactor [28]. X-ray crystallographic studies suggest that molecules inhibit enzymes by coordinating with active site meatal ion (Co^2+^). Isatin-linked hydrazone B, C [29,30], pyrazole D [31] and compound A (Figure 1) showed antimicrobial activity by inhibiting *Ec*MetAP1. These active pharmacophores inspired us develop new isatin derivatives.

Herein, we describe the synthesis and evaluation of Isatin-pyrazole hydrazone conjugates (**PS1–14**) as novel inhibitors of bacterial MetAPs. **PS3**, **PS6**, **PS9** and **PS11** showed potent and selective inhibition of microbial MetAPs compared to the human enzyme. Encouraged by the enzyme inhibition, we determined MIC, MBC, disk-diffusion assay and combination study on these four compounds against various microbes. Based on these experiments, **PS9** was identified as the most effective antibacterial compound against the standard and environmental MDR bacterial isolates. Cytotoxicity and hemolysis assays were carried out to assess the effect of the **PS9** on human embryonic kidney and hRBCs. Together, the data described in this manuscript suggest that Isatin hydrazone conjugate (**PS9**) is good a bacterial MetAP inhibitor with high selectivity and antibacterial efficacy in environmental MDR bacterial isolates.

## 2. Results

### 2.1. Chemistry

The pyrazole linked Isatin hydrazone conjugates were synthesized using the general synthetic approach as shown in Figure 1. The method features an easily accessible methyl ketone; 3-bromo acetophenone **1** underwent Claisen condensation with diethyl oxalate to form enolate, which was further reacted with diethyl oxalate **2** in ethanol to produce β-keto ester **3**. The diketo ester **3** was refluxed with hydrazine hydrate to underwent cyclo-condensation in presence of glacial acetic acid to form pyrazole ester; ethyl 5-(3-bromophenyl)-1H-pyrazole-3-carboxylate **4**. The pyrazole ester **4** was further treated with hydrazine hydrate, underwent nucleophilic substitution followed by condensation to form 5-(3-bromophenyl)-1H-pyrazole -3-carbohydrazide **5**. In a separate set of reaction, variously substituted anilines (**6a–n)** bearing electron-withdrawing and electron-donating groups reacted with chloral hydrate and hydroxylamine hydrochloride to form corresponding isonitrosoacetanilide (**7a–n)** which on cyclization yielded substituted Isatins (**8a–n)**. Finally, substituted Isatins (**8a–n)** were reacted with 5-(3-bromophenyl)-1H-pyrazole-3-carbohydrazide **5** to form the corresponding Isatin-pyrazole hydrazones (**PS1–14**). All the intermediates and final compounds were characterized using multi-spectroscopic techniques such as FT-IR, ^1^H, ^13^C NMR and mass spectrometry. Briefly, the appearance of peaks at 3350 and 3106 cm^−1^ in the FT-IR pertaining to asymmetric and symmetric stretching of N*H*_2_ group of hydrazide indicated the conversion of ester to hydrazide. The formation of hydrazine was indicated with the appearance of a characteristic peak corresponding to N*H* and C=O stretching at 3188–3376 and 1683–1737 cm^−1^ in the IR spectra of Isatin-pyrazole hydrazones. The ^1^H NMR spectra of final compounds exhibited characteristic sharp singlet for the -NH proton pyrazole ring in the range of 12.5–14 ppm. Two singlets corresponding to NH protons belonging to hydrazones and Isatin groups appeared in the range 11.25–12.50 ppm and 10.00–11.50 ppm, respectively, indicated the formation of Isatin-pyrazole hydrazones. In the ^13^C NMR spectra, the peaks for C1, C4 and C5 carbon atoms of pyrazole ring appeared in the range 120.72–148.72 ppm, respectively. Moreover, peaks corresponding to Isatin carbonyl functionality, and the carbonyl group of hydrazones appeared in the range 162.94–165.02 ppm and 158.73–160.23 ppm, respectively. The mass spectra of all the compounds were found to be in agreement with the calculated values thus confirming the formation of the desired compounds. All the final compounds were confirmed to have ≥90% purity before performing any biological activity.

### 2.2. Pharmacological Evaluation

#### 2.2.1. Drug-Likeliness Assessment

After the careful assessment of all the **14** conjugates, all were found to fulfil the Ro5 norms. The physicochemical properties of all conjugates are provided in Appendix A. All the Isatin-pryazole hydrazones under study were found to contain less than 15 rotable bands. Carcinogenicity of compounds was checked using carcinopred-EL; we found that all compounds are non-carcinogenic in nature.

#### 2.2.2. Inhibition and Kinetic Studies of Methionine Aminopeptidases

All the Isatin-pyrazole hydrazone conjugates were subjected to biochemical assay with the purified MetAPs from different organisms. The compounds which induced ≥40% inhibition against enzymes at 10 µM concentration (Appendix A) were further examined for their ***K****_i_* value. *Mt*MetAP1c enzyme showed ≥40% inhibition with eight compounds and additional with **PS3** compound; a total of nine compounds were taken further to find their ***K****_i_* values (Table 1). *Ef*MetAP1a enzyme produced ≥40% inhibition for three compounds and additional with **PS1**, **PS6** and **PS9** compounds; a total of six compounds were evaluated for their ***K****_i_* values (Table 1). *Sp*MetAP1a enzyme got ≥40% inhibition for five compounds and additional with **PS3** and **PS6** compounds; total seven compounds ***K****_i_* values were determined (Table 1). *Hs*MetAP1b enzyme experienced much less inhibition or no inhibition against these compounds confirming their selectivity.

#### 2.2.3. Growth Inhibition Assay

To confirm their antimicrobial efficacy, all the compounds were subjected to in vitro antimicrobial activities. Initially, 250 µg/mL as the single highest concentration was used to start in vitro screening to classify potent compounds and those compounds which are ineffective at this concentration were excluded. Most of the compounds exhibited significant inhibition against *K. pneumoneae, P. aeruginosa* and *E. coli* bacterial strains. However, little significant inhibition was observed against *E. faecalis* after the treatment with any of the test compounds. Compounds **PS1, PS13** and **PS14** did not show any significant antibacterial potential, while compounds **PS5, PS7** and **PS10** showed selective inhibition against tested bacterial strains. Furthermore, **PS3, PS6, PS9** and **PS11** were found to have good to moderate activity against a variety of bacterial species*. B. subtilis* showed 100% growth inhibition when treated with the compounds **PS6, 7, 9** and **11**. Compound **PS3** showed 97% inhibition in the case of *K. pneumoniae*, however **PS6** produced more than 70% inhibition for all the tested strains except *S. aureus* (61%). Compounds **PS9** and **PS11** were identified as potent antimicrobials as ≥94% growth inhibition was observed against all the tested strains. Some of the compounds were not able to inhibit the growth of tested strains and therefore were excluded from further studies. Ciprofloxacin (CIP) and ampicillin (AMP) were used as standard drugs. The activity results are presented in Table 2.

#### 2.2.4. Minimum Inhibitory Concentration (MIC)

Based on their growth inhibitory potential, the selected compounds (**PS3**, **6**, **9** and **11**) were further evaluated as possible antimicrobials. Compound **PS9** demonstrated good to moderate antibacterial potential, having an MIC value of 8 µg/mL against *P. aeruginosa* and *K. pneumonia* whereas against *E. coli* the MIC value was 16 µg/mL. **PS9** exhibited MIC values of 8 µg/mL against *E. faecalis* and 32 µg/mL against *S. aureus* and *B. subtilis.* The antibacterial activity of **PS6** was observed with MICs of 8 µg/mL against *B. subtilis* and 16 µg/mL against *E. faecalis* and *S. aureus.*
**PS3** and **PS11** also produced mild to moderate antibacterial activity, with **PS11** having a MIC of 32 µg/mL for *E. faecalis* along with *K. pneumoniae* and *P. aeruginosa,* and **PS3** having an MIC of 32 µg/mL against *K. pneumoniae* and *P. aeruginosa* (Table 3). Compound **PS11** showed no substantial inhibitory activity against *B. subtilis* with MIC values greater than 256 µg/mL. Overall, the results suggested that the compounds **PS6** and **PS9** showing lowest MICs against selective bacterial strains are the lead inhibitors with potent and selective bacterial MetAP inhibition properties.

#### 2.2.5. Minimum Bactericidal Concentration (MBC)

The MBC value demonstrates the lowest level of antimicrobial agent resulting in microbial death. The antibacterial activity of a compound was measured as the ratio of MBC to MIC. The compound is classified as bactericidal if the MBC/MIC ratio is *≤*2. Whereas, if the ratio lies between 2 to 16, it will be considered as bacteriostatic in nature. For this, compounds **PS3, PS6, PS9**, and **PS11** were used to determine MBCs against aforesaid standard isolates. **PS3** was found to be bactericidal against *E. coli, B. subtilis* and *S. aureus* as the ratio of MBC/MIC lies between 1 to 2. However, isolates *E.*
*faecalis, K. pneumonia* and *B. subtilis* had identical MBC/MIC values i.e., 2 for the compound **PS6**; hence, **PS6** also reflected bactericidal nature against these strains. MBC/MIC value was observed *≤2* against *P. aeruginosa* and *K. pneumoniae* in case of compound **PS9**, whereas **PS11** was found to be bactericidal in nature against *E. faecalis* and *K. pneumoniae* (Table 4). In conclusion, compounds were found to be selectively bactericidal to bacteriostatic in nature.

#### 2.2.6. Disk Diffusion Assay

The antibacterial efficacy of the test compounds **PS3**, **PS6**, **PS9** and **PS11** was determined using the disk diffusion assay on Muller Hinton Agar (MHA) at concentrations corresponding to ½ MIC, MIC and 2MICs. On treatment of **PS3** and **PS9,** clear Zone of Inhibition (ZOI) ranging from 06 to 10 mm was measured around the disk with all six bacterial cultures. However, on treatment of **PS6** and **PS11,** ZOI was measured around the disk between 07 to 11 mm. Zone of inhibition (mm) measured around the disk of various concentration of the tested compounds are given in Appendix A.

#### 2.2.7. Combination Study of the Selected Compounds

**PS6** and **PS9** were carried forward for combination study with CIP and AMP against *E. faecalis, P. aeruginosa, B. subtilis, K. pneumoniae S. aureus* and *E. coli* bacterial strains to see if they had a synergistic antimicrobial effect. The results showed that **PS6** has a significant increase in antibacterial effectiveness against *E. coli, K. pneumoniae, S. aureus E.*
*faecalis* and *P. aeruginisa* strains in combination with AMP, i.e., in a synergistic mode of interaction, but **PS6** has an indifferent mode of interaction with AMP against *B. subtilis*, while CIP had a synergistic effect only against *E. faecalis and E. coli* (Table 5 and Table 6).

Compound **PS9** with **AMP** had a synergistic impact against *K. pneumoniae E. faecalis*, *P. aeruginosa, S. aureus* and *E. coli*, whereas in combination with CIP has a synergistic effect against *S. aureus* and all tested Gram-negative isolates. In addition to this, **PS9** has an indifferent mode of interaction with a combination of CIP against *E. faecalis,* while a synergistic mode of interaction was observed when combined with AMP. The results showed a synergistic effect of both the compounds (**PS6** and **PS9**) with AMP and CIP against all Gram-negative strains (except *K. pneumonia*) (Table 5 and Table 6).

These results suggest that compound **PS9** has good synergistic effect (i.e., many fold decrease in MIC values of compound) with standard drugs (AMP and CIP), which may be useful for treating resistant bacterial strains using combination therapy.

#### 2.2.8. Growth Kinetics

A growth kinetics study was performed to determine the effect of the lead compound **PS9** on the growth of the test organisms. The growth kinetics of two Gram-positive and two Gram-negative strains were studied in the presence of various concentrations of **PS9.** The results revealed control bacterial cell growth curve with a noticeable lag, exponential, stationary and decline phases. In the case of *E. coli* and *P. aeruginosa* culture, there was no growth observed up to 24 h at 2MIC and MIC, whereas at ½ MIC, for *E. coli* and *P. aeruginosa* culture log phase was delayed since the log phase was observed after 6 and 8 h of inoculation respectively (Figure 2). In the cultures of *E. faecalis* and *K. pneumoniae,* no significant growth was found at MIC up to 10 h, although no growth was observed in both cultures at 2MIC concentration till 24 h. Compound **PS9** delayed the growth up to 6 h even at ½ MIC concentration in all the tested strains and found most effective against *E. coli* and *P. aeruginosa*. Thus, the result showed that compound **PS9** displayed strong antibacterial properties against the tested strains.

#### 2.2.9. Time Kill Curve Assay

The bacteriostatic or bactericidal nature of **PS9** against *P. aeruginosa, E. faecalis, K. pneumoniae* and *E. coli* was also determined by Time-kill curve study. Two different concentrations equivalent to MIC and 4MIC were used to determine dose dependent response of the test compound and ampicillin was used as the reference. The study found that the regular decreases in the colony-forming unit (CFU) with the time interval of up to <1 h were detected up to 16 h, indicating killing activity. There was a significant decline in log_10_ CFU/mL for tested strains (*E. coli, E. faecalis, K. pneumoniae* and *P. aeruginosa)* with time at the MIC concentration, but no 100% eradication of the bacterial population was seen till 16 h. At higher concentrations (4MIC), significant eradication of *E. coli, E. faecalis, K. pneumoniae* and *P. aeruginosa* bacterial cells was seen after 16 h (Figure 3). Based on the growth curve and time kill curve assays, **PS9** was found to be bactericidal against the tested bacterial strains.

#### 2.2.10. In Vitro Study of PS9 on Environmental Resistant Strains

##### MIC of PS9 Using Environmental Resistant Strains

We determined the MIC value of **PS9** against the different environmental isolates and compared it with standard drugs ampicillin (AMP) and cefotaxime (CTX). Results showed that the MIC of the **PS9** against tested isolates was lower, equivalent, and even higher than the standard drugs. There was a significant decrease in MIC against strains isolated from the lake, as the MIC of the compound in the case of isolate HK1 is 16 times lower than standard antibiotic AMP and 8 times lower than third-generation drug CTX. Isolates HK32, HK41, HK43 and HK91 exhibited good to moderate inhibition affinity as compared to standard drugs. **PS9** has a MIC value of 2 µg/mL, three times less than AMP and CTX in the instance of HK91 (Table 7). LG20 has a MIC of less than 1 µg/mL, which is 32 times lower than AMP and CTX. Even for LG1 and JST68, **PS9** showed judicious MIC values for the test compound. Some extremely resistant strains, including LG71, JST71, SD5 and SD6, had higher MIC values for the compound (1024 µg/mL).

##### MBC of Environmental Resistant Strains

**PS9** exhibited a bactericidal nature against *Enterobcter cloacae* (HK32) since its MIC and MBC values are the same. The MBC and MIC ratio is 2 for the *Exiguobacterium mexicanum* (HK43) and *Enterobacter cloacae* (HK41) isolates indicate that **PS9** has bactericidal nature against these environmental isolates. The lowest MBC value 16 µg/mL was observed against *Enterobacter* sp. (LG20) and *Ralstonia* sp. (HK91) (Table 3 and Table 6). In brief PS9 selectively showed bactericidal as well as bacteriostatic nature against environmental isolates.

### 2.3. Evaluation of In Vitro and In Vivo Toxicity

#### 2.3.1. Hemolytic Activity

Spectrophotometric assessment was conducted to assess the impact of **PS9** on human red blood cells (*h*RBCs) and HEK293 cells. Up to 31.25 µg/mL concentration, no significant lysis in RBCs was observed by the **PS9**, whereas 1% Triton-X 100 treated RBCs showed 100% hemolysis confirming non-toxic nature of PS9 (Figure 4a).

#### 2.3.2. Cytotoxicity Assay

Compound **PS-9** did not have any effect on the viability of human embryonic kidney (HEK293) cells because **PS9** is non-cytotoxic to human cells in the concentration range tested. At 100 µg/mL concentration of PS9, beyond 90% HEK293 cells were sustainable. More than 75 percent cells were found to be viable in the presence (up to 200 µg/mL) of the compound **PS9** (Figure 4b).

##### In Vivo Toxicological Assessment of PS9 in *G. mellonella* Larvae

Larvae administered a range of PS9 concentrations (100, 250, 500, 750 and 1000 μg/mL) maintained 100% survival across all treatment groups for up to 5 days. PS9 appeared non-toxic to larvae at all test concentrations as larvae presented no signs of discoloration/adverse behavior and progressed into normal pupal stages (Figure 5).

##### Determination of Hemocyte Density

Hemocytes were retrieved and enumerated from larvae inoculated with **PS9** (500 and 1000 μg/mL) over a 24 h period at 37 °C. Hemocyte levels were assessed to ascertain whether **PS9** elicited an immune response in larvae. Hemocyte densities were largely comparable with untreated control samples, except in the case of 500 μg/mL **PS9** treatment at the 24 h point (Figure 6).

##### Assessment of In Vivo Antibacterial Efficacy of PS9 in *G. mellonella*

*G. mellonella* larvae were infected with *P. aeruginosa* PAO1 (Gram-negative) and *S. aureus* 33,591 (Gram-positive) with cell densities that resulted in approximately 50% and 80% larval death after 24 h (i.e., LD_50_ and LD_80_). Larvae infected with *S. aureus* (~2 × 10^9^ CFU/mL and ~1 × 10^9^ CFU/mL) were incubated at 37 °C for 30 min and treated with the highest tested dose of **PS9** (1000 μg/mL). Larval survival was observed for up to 72 h. This was repeated using *P. aeruginosa* (~3 × 10 CFU/mL and ~1.5 × 10 CFU/mL) (Figure 7A,B).

The results presented here indicate that PS9 is non-toxic to *G. mellonella* larvae and does not result in death, melanization or any developmental alterations at doses of up to 1000 mg/mL (Figure 6). Administration of PS9 to larvae does not result in a change in the hemocyte (immune cell) density over 24 h indicating the lack of an immune response (Figure 7). Infection of larvae with *S. aureus* or *P. aeruginosa* results in death over three days—the administration of PS9 does not delay this although in the case of *P. aeruginosa* infected larvae there may be a small protective effect in larvae infected with 1.5 × 10^9^ CFU/mL at 48 and 72 h (Figure 7B).

## 3. Discussion

For many years, antimicrobial resistance (AMR) has posed a serious threat to public health because bacteria have always been able to acquire, adopt and evolve antibiotic resistance mechanisms [32,33]. Antibiotic resistance is one of the top ten threats to global health as per the reports of the World Health Organisation (WHO). As a result, in this study, Isatin-pyrazole hydrazone conjugates (**PS1–14**) were synthesized and subjected to in vitro inhibition against four different methionine aminopeptidases. MetAP removes the ***N***-terminal methionine from newly synthesized peptide from protein translation. Several of these 14 Isatin-pyrazole hydrazone conjugates inhibited MetAPs from *Mycobacterium tuberculosis (Mt)*, *Enterococcus faecalis (Ef)* and *Streptococcal pneumoniae (Sp)* selectively in with *K*_i_’s in nanomolar concentrations. One of the compounds (PS9) inhibited the *Mt*MetAP1c and *Sp*MetAP1a with the *K*_i_ of 310 nM and 370 nM, respectively, and no inhibition against the human counterpart (Table 1); surprisingly PS9 showed very less inhibition towards *Ef*MetAP. Biochemical and structural studies revealed the mechanism of selectivity.

*Mt*MetAP1c showed highest inhibition towards compounds compared with the remaining enzymes. Isatin-pyrazole hydrazone conjugates, most of the modifications were added to the 2-oxoindolin-3-ylidene so without the addition of any modification (PS1) showed inhibition of 2.3 µM(Ki) against *Mt*MetAP1c. We added methyl group in the 5th position (PS02, Ki is 4.37 µM) or bromine (PS5 Ki is 3.14 µM), or chlorine (PS8) it showed a decrease in the inhibition but if we added fluorine group to this 5th position (PS9, Ki is 0.31 µM) or methoxy group (PS11, Ki is 1.61 µM), it showed an increase in the inhibition. When we added additional groups to the 6th position like chlorine (PS3, Ki is 2.32 µM) or methoxy group (PS12, Ki is 0.61 µM) or nitro (PS14, Ki is 1.27 µM), it showed an increase in the inhibition, but surprisingly, addition of fluorine group to the same position (PS7) shown a decrease in the inhibition (Appendix A). Adding modification at 7th positions with methyl (PS4) or Nitro (PS10) or chlorine (PS13) showed a decrease in inhibition in all MetAPs. Adding fluorine to the 5th position (PS09, Ki is 0.31 µM) along with the 6th position chlorine (PS6, Ki is 0.74 µM), has shown a decrease in the inhibition compared with PS9 and an increase in the inhibitions compared with PS3 (Appendix A). Surprisingly, ***Ef***MetAP1a showed the highest inhibition with the addition of fluorine to the 5th position along with the addition of chlorine to the 6th position (PS6, Ki is 1.65 µM) compared with 2-oxoindolin-3-ylidene so without the addition of any modification (PS1, Ki is 2.65 µM) and remaining all compounds shown either of the similar inhibition or decrease in the inhibition compared with PS1, which has no modifications of 2-oxoindolin-3-ylidene (Appendix A). ***Sp***MetAP1a showed an increase in the inhibition compared with PS1 (Ki is 4.48 µM), which has no modifications of 2-oxoindolin-3-ylidene, but the best inhibition was showed by adding the fluorine group to this 5th position (PS9, Ki is 0.37 µM) or nitro group to the 6th position (PS14, Ki is 0.5 µM) (Appendix A).

In preliminary screening, four compounds of this series, i.e., PS3, PS6, PS9 and PS11, were noticed to be selective growth inhibitors against *B. subtilis, E. faecalis, S. aureus, P. aeroginosa, K. pneumoniae* and *E. coli*. Minimum inhibitory concentration (MIC), Minimum bactericidal concentration (MBC), disk diffusion assay, growth curve and time-kill curve experiments were used to confirm these compounds have growth inhibitory properties [26]. The bactericidal activity of the PS9 against selected bacterial strains as demonstrated by various antibacterial investigations strongly supports the view that PS9 compound should be used as the lead compound. To avoid toxicity and other adverse effects associated with large doses of single-drug treatment, a synergistic combination of two or more pharmaceuticals is suggested, since low doses of drugs have no or mild side effects, and, moreover, using numerous targets boosts the treatment [34,35,36]. Except for *K. pneumonia,* PS9 showed a synergistic mode of interaction with AMP or CIP against all tested Gram-negative strains. Furthermore, we studied the antibacterial efficacy of PS9 against 21 multidrug-resistant strains. Though our lead compound PS9 is found to have a decreased MIC values against *Enterobacteriacea* sp (HK1), *Enterobacter cloacae* (HK32 and HK41), *Exiguobacterium mexicanum* (HK43), *Ralstonia* sp. and *E. coli* in comparison to standard drugs ampicillin (AMP) and third generation cephalosporins cefotaxime. Selectively, PS9 exhibited bactericidal activity against environmental MDR strains, i.e., *Enterobcter clocacae* (HK32), *Exiguobacterium mexicanum* (HK43) and *Enterobacter cloacae* (HK41), as MBC/MIC values lie between 1–2 [37,38]. Compound PS9, cytotoxicity tests revealed that it could be employed as a promising antibacterial agent against certain standard as well as MDR bacterial isolates because PS9 showed non-toxicity towards HEK293 cells and human red blood cells up to 31.25 µg/mL and 200 µg/mL, respectively, but specifically inhibits bacterial cells. Additionally, PS9 is non-toxic to *G. mellonella* larvae and does not result in death, melanization or any developmental alterations at doses of up to 1000 mg/mL.

## 4. Materials and Methods

### 4.1. Chemistry

All the required chemicals were purchased from Merck, GLR, Sigma-Aldrich and were used without further purification. Pre-coated Merck Silica gel 60 F_254_ TLC aluminum sheets were used under UV light and I_2_ vapor staining was used for visualization of spots. Purification was performed by column chromatography using Merck 230–400 mesh silica gel. Digital BÜCHI melting point apparatus (M-560) was used for the measurement of melting points and were uncorrected. IR spectra were recorded, and only major peaks are reported in cm^−^^1^ using Agilent Cary 630 FT-IR spectrometer. DMSO-*d_6_* (as solvent) with TMS as an internal standard on Bruker Spectrospin DPX-300 spectrometer at 300 MHz and 75 MHz were used for ^1^H and ^13^C NMR spectra respectively. s (singlet), d (doublet), t (triplet), m (multiplet) or brs (broad) were labelled as splitting pattern. The chemical shift values of ^1^H NMR were reported in parts per million (ppm) relative to residual solvent (DMSO-*d_6_*, *δ* 7.26) and ^13^C NMR chemical shifts (*δ*) of ^13^C NMR were reported in ppm relative to (DMSO-*d_6_*, *δ* 77.16) and Hertz (Hz) is the unit of coupling constants (*J*). Agilent Quadrupole-6150 LC/MS spectrometer was used for recording of mass spectra. Synthesis protocols and spectral characterization is discussed below:

#### 4.1.1. Synthesis of Ethyl 4-(3-Bromophenyl)-2,4-Dioxobutanoate (**3**)

In a dried RB flask, Na metal (0.09 g, 3.70 mmol) was cautiously added portion-wise to anhyd. EtOH (4.4 mL) under stirring until all the Na had reacted. Ethyl oxalate 2 (0.60 g, 4.07 mmol) and then the ketone 1 (3.70 mmol) were added sequentially. The solution was stirred at room temperature for 1 h and then slowly poured into ice. To this cold mixture, 2 M HCl solution was added and the resulting suspension was filtered, and the collected solid was dried in air [39].

#### 4.1.2. Synthesis of Ethyl 3-(3-Bromophenyl)-1H-Pyrazole-3-Carboxylate (**4**)

To a mixture of diketoester 3 (1 g. 3.3 mmol) in acetic acid (10 mL), hydrazine hydrate (0.247 g. 4.95 mmol, 1.5 equiv.) was added and the reaction mixture was refluxed for 4 h. The reaction was confirmed by TLC, the mixture was concentrated, cooled and poured into crushed ice. The obtained solid was filtered and dried to afford a light-yellow solid material (0.75 g. 75%) [39].

#### 4.1.3. Synthesis of 3-(3-Bromophenyl)-1H-Pyrazole-3-Carbohydrazide (**5**)

In a dried RB flask, a solution of pyrazole ethyl ester (ethyl 3-(3-bromophenyl) isoxazole-5-carboxylate) 4 (1.0 equiv.) in ethanol was added hydrazine hydrate (5.0 equiv.) dropwise and the reaction mixture was refluxed overnight at 80 °C. After completion of the reaction, the reaction mixture was cooled to room temperature and added ice-cold water. The solid precipitate obtained was filtered and washed with cold ethanol. The solid product was dried under vacuo to afford the desired product [40].

#### 4.1.4. Synthesis of Substituted Isonitrosoacetanilides (**7a**–**n**)

To a solution of substituted aniline (6a–n) (4.29 mmol) in a dry round bottom flask was added water (23 mL) and 2 N HCl (1.72 mL). After complete dissolution of substituted aniline was then added in the order: anhyd. sodium sulphate (28.35 mmol), hydroxylamine hydrochloride (15.03 mmol) and chloral hydrate (5.58 mmol). The reaction mixture was then allowed to stir overnight at 55 °C and cooled to room temperature after completion of the reaction as monitored by TLC. The crude was filtered under vacuo, washed with distilled water to afford substituted isonitrosoacetanilides (7a–n) which were used to form Isatin derivatives without any further purification [40].

#### 4.1.5. Synthesis of Substituted Isatin (**8a**–**n**)

In a two-neck round bottom flask, 1.08 mL of concentrated sulfuric acid was warmed to 50 °C followed by addition of 1.52 mmol of dry isonitrosoacetanilide (7a–n) at such a rate to keep the temperature of the reaction vessel between 60–70 °C. The reaction mixture was heated to 80 °C for 15–20 min to complete the reaction as monitored by TLC. Finally, the reaction mixture was cooled to room temperature and poured onto ice water over three times its volume. The solid precipitate was filtered with suction, washed with water until sulfuric acid was completely removed, and dried to yield substituted Isatin (8a–n) with high purity [40].

#### 4.1.6. Synthesis of Substituted 3-(3-Bromophenyl)-N′-(2-Oxoindolin-3-Ylidene) -1H-Pyrazole-3-Carbohydrazide PS (**1**–**14**)

In a dried RB flask, pyrazole hydrazide (3-(3-bromophenyl)isoxazole-5-carbohydrazide) 5 (1.0 equiv.) dissolved in ethanol was added appropriate Isatin derivatives 8a–n followed by few drops of glacial acetic acid and the reaction mixture was refluxed for overnight at 80 °C. After completion of the reaction, the reaction mixture was cooled to room temperature and added ice-cold water. The solid precipitate obtained was filtered through a funnel and washed with ethanol. The solid product was dried under vacuo to afford the desired product (PS1–14) [40].

#### 4.1.7. (E)-3-(3-Bromophenyl)-N′-(2-Oxoindolin-3-YLIDENE)-1H-Pyrazole-5-Carbohydrazide (**PS1**)

Light yellow solid; yield 77%; R_f_ = 0.75 (Ethylacetate: Hexane = 50:50); mp: 218–221 °C; FT-IR (cm^−1^): 3298, 3033, 2842, 1610, 1546, 1499, 1320, 1152; ^1^H NMR (500 MHz, DMSO- *d_6_*) (*δ*, ppm): 13.88 (s, 1H, N-*H,* pyrazole ring), 11.32 (s, 1H, N*H*, hydrazone), 10.09 (s, 1H, N-*H,* Isatin ring), 7.57 (d, *J* = 7.4 Hz, 1H, Ar-*H*), 7.42 (d, *J* = 2.0 Hz, 1H, Ar-*H*), 7.34 (ddd, *J* = 16.8, 8.4, 1.6 Hz, 3H, Ar-*H*), 7.10–7.06 (m, 1H, Ar-*H*), 6.93 (dd, *J* = 8.0, 3.9 Hz, 2H, Ar-*H*),6.92 (s, 1H, Ar-*H*);^13^C NMR (125 MHz, DMSO-*d*_6_) (*δ*, ppm): 164.8, 163.6, 145.9, 132.73, 130.9, 130.1, 129.9, 129.2, 127.2, 124.5, 124.1, 123.7; ESI-MS (*m*/*z*) calcd. for C_18_H_12_BrN_5_O_2_: 409.01 Found: 410.25 [M+H]^+^

#### 4.1.8. (E)-3-(3-Bromophenyl)-N′-(5-Methyl-2-Oxoindolin-3-Ylidene)-1H-Pyrazole-5-Carbohydrazide (**PS2**)

Yellow solid; yield 77%; R_f_ = 0.73 (Ethylacetate: Hexane = 50:50); mp: 209–213 °C; FT-IR (cm^−1^): 3349, 3212, 2953, 2853, 1548, 1473, 1335, 1160; ^1^H NMR (500 MHz, DMSO-*d_6_*) (*δ*, ppm): 13.93 (s, 1H, N-*H*, pyrazole ring), 11.34 (s, 1H, N*H*, hydrazone), 10.08 (s, 1H, N-*H,* Isatin ring), 7.57 (d, *J* = 7.4 Hz, 1H, Ar-*H*), 7.42 (d, *J* = 2.0 Hz, 1H, Ar-*H*), 7.34 (ddd, *J* = 16.8, 8.4, 1.6 Hz, 3H, Ar-*H*), 7.10–7.06 (m, 1H, Ar-*H*), 6.93 (dd, *J* = 8.0, 3.9 Hz, 2H, Ar-*H*), 2.70 (s, 3H, C*H_3_*). ^13^C NMR (125 MHz, DMSO-*d*_6_) (*δ*, ppm): 164.0, 162.1, 143.8, 140.9, 137.7, 134.0, 133.4, 132.9, 131.3, 130.1, 128.7, 125.6, 121.8, 121.4, 20.4; ESI-MS (*m*/*z*) calcd. for C_19_H_14_BrN_5_O_2_: 423.0331 Found: 424.10 [M+H]^+^.

#### 4.1.9. (E)-3-(3-Bromophenyl)-N′-(6-Chloro-2-Oxoindolin-3-Ylidene)-1H-Pyrazole-5-Carbohydrazide (**PS3**)

Light yellow solid; yield 77%; Rf = 0.71 (Ethylacetate: Hexane = 50:50); mp: 182–185 °C; FT-IR (cm^−1^): 3328, 3032, 1742, 1549, 1469,1328,1154; 1H NMR (500 MHz, DMSO-d6) (δ, ppm): 13.90 (s, 1H, N-H, pyrazole ring), 11.13 (s, 1H, NH, hydrazone), 10.10 (s, 1H, N-H, Isatin ring), 7.42 (d, *J* = 1.8 Hz, 1H, Ar-H), 7.32 (dd, *J* = 8.3, 1.8 Hz, 2H, Ar-H), 7.10 (d, *J* = 1.3 Hz, 1H, Ar-H), 6.92 (d, *J* = 8.2 Hz, 3H, Ar-H), 6.84 (d, *J* = 8.5 Hz, 1H, Ar-H);13C NMR (125 MHz, DMSO-d6) (δ, ppm): 164.0, 162.3, 146.0, 137.8, 133.4, 133.0, 131.3, 131.0, 130.3, 129.5, 128.7, 124.3, 124.2, 121.8; ESI-MS (*m*/*z*) calcd. for C18H11BrClN5O2:442.9785 Found: 443.60 [M+H]+.

#### 4.1.10. (E)-3-(3-Bromophenyl)-N′-(5,7-Dimethyl-2-Oxoindolin-3-Ylidene)-1H-Pyrazole-5-Carbohydrazide (**PS4**)

Yellow solid; yield 76%; R_f_ = 0.69 (Ethylacetate: Hexane = 50:50); mp: 257–260 °C; FT-IR (cm^−1^): 3323, 3243, 3070, 2917, 1738 1490, 1332, 1170; ^1^H NMR (500 MHz, DMSO-*d_6_*) (*δ*, ppm): 13.93 (s, 1H, N-*H*, pyrazole ring,), 11.13 (s, 1H, N*H*, hydrazone), 10.10 (s, 1H, N-*H*, Isatin ring), 7.42 (d, *J* = 8.1 Hz, 1H, Ar-*H*), 7.41 (d, *J* = 2.0 Hz, 1H, Ar-*H*), 7.31 (dd, *J* = 8.3, 2.0 Hz, 1H, Ar-*H*), 7.10 (dd, *J* = 8.1, 1.8 Hz, 3H, Ar-*H*), 6.85 (d, *J* = 1.8 Hz, 1H, Ar-*H*), 2.82 (s, 3H, C*H_3_*), 2.74 (s, 3H, C*H_3_*). ^13^C NMR (125 MHz, DMSO-*d*_6_) (*δ*, ppm): 164.1, 163.7, 145.8, 137.4, 130.9, 130.2, 130.1, 129.3, 129.0, 124.4, 124.1, 122.6, 106.9, 105.2, 99.6, 56.6, 56.1; ESI-MS (*m*/*z*) calcd. for C_20_H_16_BrN_5_O_2_: 437.0487 Found: 438.01 [M+H]^+^.

#### 4.1.11. (E)-N′-(5-Bromo-2-Oxoindolin-3-Ylidene)-3-(3-Bromophenyl)-1H-Pyrazole-5-Carbohydrazide (**PS5**)

Yellow solid; yield 79%; R_f_ = 0.70 (Ethylacetate: Hexane = 50:50); mp: 230–233 °C; FT-IR (cm^−1^): 3218, 2997, 1738, 1588, 1523, 1337, 1162; ^1^H NMR (500 MHz, DMSO-*d*_6_) (*δ*, ppm): 13.77 (s, 1H, N-*H*, pyrazole ring), 11.45 (s, 1H, N*H*, hydrazone), 10.11 (s, 1H, N-*H*, Isatin ring), 7.56 (d, *J* = 8.1 Hz, 2H, Ar-*H*), 7.41 (d, *J* = 2.0 Hz, 1H, Ar-*H*), 7.31 (dd, *J* = 8.3, 2.0 Hz, 2H, Ar-*H*), 7.10 (dd, *J* = 8.1, 1.8 Hz, 2H, Ar-*H*), 6.94 (d, *J* = 1.8 Hz, 1H, Ar-*H*); ^13^C NMR (125 MHz, DMSO-*d*_6_) (*δ*, ppm): 163.0, 161.2, 145.7, 132.4, 131.3, 130.9, 130.1, 129.3, 129.2, 124.5, 124.3, 124.0; ESI-MS (*m*/*z*) calcd. for C_18_H_11_Br_2_N_5_O_2_: 486.9279 Found: 487.20 [M+H]^+^.

#### 4.1.12. (E)-3-(3-Bromophenyl)-N′-(6-Chloro-5-Fluoro-2-Oxoindolin-3-Ylidene)-1H-Pyrazole-5-Carbohydrazide (**PS6**)

Light yellow solid; yield 79%; R_f_ = 0.71 (Ethylacetate: Hexane = 50:50); mp: 191–194 °C; FT-IR (cm^−1^): 3315, 2946, 2838, 1738, 1594, 1494, 1363, 1154; ^1^H NMR (500 MHz, DMSO-*d*_6_) (*δ*, ppm): 13.87 (s, 1H, N-*H*, pyrazole ring), 11.20 (s, 1H, N*H*, hydrazone), 10.08 (s, 1H, N-*H*, Isatin ring), 7.41 (d, *J* = 1.9 Hz, 1H, Ar-*H*), 7.38 (s, 1H, Ar-*H*), 7.31 (dd, *J* = 8.3, 1.8 Hz, 1H, Ar-*H*), 7.14 (d, *J* = 7.8 Hz, 1H, Ar-*H*), 6.92 (d, *J* = 8.2 Hz, 2H, Ar-*H*), 6.81 (d, *J* = 7.9 Hz, 1H, Ar-*H*);^13^C NMR (125 MHz, DMSO-*d*_6_) (*δ*, ppm): 163.50, 163.46, 153.5, 152.3, 145.9, 130.9, 130.1, 129.1, 124.5, 124.0, 119.8, 115.08, 115.07, 112.9; ESI-MS (*m*/*z*) calcd. for C_18_H_10_BrClFN_5_O_2_: 460.96 Found: 461.01 [M+H]^+^.

#### 4.1.13. (E)-3-(3-Bromophenyl)-N′-(6-Fluoro-2-Oxoindolin-3-Ylidene)-1H-Pyrazole-5-Carbohydrazide (**PS7**)

Yellow solid; yield 81%; R_f_ = 0.73 (Ethylacetate: Hexane = 50:50); mp: 228–230 °C; FT-IR (cm^−1^): 3388, 3272, 2950, 1738, 1548, 1484, 1315; ^1^H NMR (500 MHz, DMSO-*d*_6_) (*δ*, ppm): 13.81 (s, 1H, N-*H*, pyrazole ring), 11.25 (s, 1H, N*H*, hydrazone), 10.08 (s, 1H, N-*H*, Isatin ring), 7.41 (d, *J* = 1.9 Hz, 1H, Ar-*H*), 7.38 (s, 1H, Ar-*H*), 7.31 (dd, *J* = 8.3, 1.8 Hz, 2H, Ar-*H*), 7.14 (d, *J* = 7.8 Hz, 1H, Ar-*H*), 6.92 (d, *J* = 8.2 Hz, 2H, Ar-*H*), 6.81 (d, *J* = 7.9 Hz, 1H, Ar-*H*); ^13^C NMR (125 MHz, DMSO-*d*_6_) (*δ*, ppm): 164.4, 163.6, 159.1, 132.7, 132.7, 132.5, 129.9, 127.1, 126.6, 123.8, 115.5, 114.3,; ESI-MS (*m*/*z*) calcd. for C_18_H_11_BrFN_5_O_2_: 427.00 Found: 428.09 [M+H]^+^.

#### 4.1.14. (E)-3-(3-Bromophenyl)-N′-(5-Chloro-2-Oxoindolin-3-Ylidene)-1H-Pyrazole-5-Carbohydrazide (**PS8**)

Yellow solid; yield 76%; R_f_ = 0.70 (Ethylacetate: Hexane = 50:50); mp: 217–220 °C; FT-IR (cm^−1^): 3262, 3083, 1738, 1609, 1347, 1157; ^1^H NMR (500 MHz, DMSO- *d*_6_) (*δ*, ppm): δ 13.97 (s, 1H), 11.23 (s, 1H, N-*H*, pyrazole ring), 10.08 (s, 1H, N*H*, hydrazone), 7.42 (d, *J* = 2.0 Hz, 1H, N-*H,* Isatin ring), 7.32 (dd, *J* = 8.3, 1.9 Hz, 2H, Ar-*H*), 7.31 (s, 1H, Ar-*H*), 7.22 (d, *J* = 8.2 Hz, 2H, Ar-*H*), 6.93 (d, *J* = 7.4 Hz, 2H, Ar-*H*). ^13^C NMR (125 MHz, DMSO-*d*_6_) (*δ*, ppm): 164.0, 163.1, 162.9, 159.5, 158.8, 132.7, 132.5, 131.8, 126.3, 115.7, 114.3, 106.8, 105.2, 99.6; ESI-MS (*m*/*z*) calcd. for C_18_H_11_BrClN_5_O_2_: 442.9785 Found: 443.03 [M+H]^+^.

#### 4.1.15. (E)-3-(3-Bromophenyl)-N′-(5-Fluoro-2-Oxoindolin-3-Ylidene)-1H-Pyrazole-5-Carbohydrazide (**PS9**)

Dark yellow; yield 76%; R_f_ = 0.70 (Ethylacetate: Hexane = 50:50); mp: 229–232 °C; FT-IR (cm^−1^): 3152, 3033, 2891, 1582, 1457, 1325, 1164; ^1^H NMR (500 MHz, DMSO-d_6_) (δ, ppm): δ 13.88 (s, 1H, N-H, pyrazole ring), 11.32 (s, 1H, NH, hydrazone), 10.09 (s, 1H, N-H_,_ Isatin ring), 7.57 (d, *J* = 7.4 Hz, 1H, Ar-H), 7.42 (d, *J* = 2.0 Hz, 1H, Ar-H), 7.34 (ddd, *J* = 16.8, 8.4, 1.6 Hz, 3H, Ar-H), 7.10–7.06 (m, 1H, Ar-H), 6.93 (dd, *J* = 8.0, 3.9 Hz, 2H, Ar-H); ^13^C NMR (125 MHz, DMSO-d_6_) (δ, ppm): 164.2, 163.5, 162.9, 159.6, 143.7, 140.2, 133.9, 131.8, 129.7, 125.3, 122.0, 106.9, 105.2, 99.6; ESI-MS (m/z) calcd. for C_18_H_11_BrFN_5_O_2_: 427.0080 Found: 428.00 [M+H]^+^.

#### 4.1.16. (E)-3-(3-Bromophenyl)-N′-(7-Nitro-2-Oxoindolin-3-Ylidene)-1H-Pyrazole-5-Carbohydrazide (**PS10**)

Light yellow solid; R_f_ = 0.71 (Ethylacetate: Hexane = 50:50); yield 75%; mp: 215–218 °C; FT-IR (cm^−1^): 3362, 3255, 3079, 1738, 1563, 1456, 1158; ^1^H NMR (500 MHz, DMSO-*d_6_*) (*δ*, ppm): 13.92 (s, 1H, N-*H*, pyrazole ring), 11.20 (s, 1H, N*H*, hydrazone), 10.02 (s, 1H, N-*H*,_._ Isatin ring), 7.42 (d, *J* = 7.4 Hz, 1H, Ar-*H*), 7.38 (d, *J* = 2.0 Hz, 1H, Ar-*H*), 7.32 (d, *J* = 1.6 Hz, 2H, Ar-*H*), 7.15 (d, *J* = 2.1 Hz, 1H, Ar-*H*), 6.92 (d, *J* = 8.0 Hz, 2H Ar-*H*,), 6.80 (d, *J* = 2.8.0 Hz, 1H, Ar-*H*). ^13^C NMR (125 MHz, DMSO-*d*_6_) (*δ*, ppm): 164.2, 163.4, 148.8, 145.9, 133.3, 131.8, 130.9, 130.4, 129.4, 127.0, 125.2, 124.3, 124.2, 121.7; ESI-MS (*m*/*z*) calcd. for C_18_H_11_BrN_6_O_4_: 454.0025 Found: 455.01 [M+H]^+^.

#### 4.1.17. (E)-3-(3-Bromophenyl)-N′-(5-Methoxy-2-Oxoindolin-3-Ylidene)-1H-Pyrazole-5-Carbohydrazide (**PS11**)

Reddish brown; yield 74%; R_f_ = 0.70 (Ethylacetate: Hexane = 50:50); mp: 245–248 °C; FT-IR (cm^−1^): 3379, 2971, 2839, 1738, 1607, 1585, 1460, 1330; ^1^H NMR (500 MHz, DMSO-*d*_6_) (*δ*, ppm): 13.81 (s, 1H, N-*H*, pyrazole ring), 11.20 (s, 1H, N*H*, hydrazone), 10.15 (s, 1H, N-*H*, Isatin ring), 7.42 (d, *J* = 7.4 Hz, 1H, Ar-*H*), 7.38 (d, *J* = 2.0 Hz, 1H, Ar-*H*), 7.32 (d, *J* = 1.6 Hz, 2H, Ar-*H*), 7.15 (d, *J* = 2.1 Hz, 1H, Ar-*H*), 6.92 (d, *J* = 8.0 Hz, 2H, Ar-*H*), 6.80 (d, *J* = 2.8.0 Hz, 1H, Ar-*H*), 3.82 (s, 3H, OC*H_3_*). ^13^C NMR (125 MHz, DMSO-*d*_6_) (*δ*, ppm): 164.3, 162.9, 159.3, 148.8, 133.2, 132.9, 132.7, 131.8, 126.8, 125.3, 121.6, 115.2, 114.3; ESI-MS (*m*/*z*) calcd. for C_19_H_14_BrN_5_O_3_: 439.0280 Found: 440.20 [M+H]^+^.

#### 4.1.18. (E)-3-(3-Bromophenyl)-N′-(6-Methoxy-2-Oxoindolin-3-Ylidene)-1H-Pyrazole-5-Carbohydrazide (**PS12**)

Dark brown solid; yield 81%; R_f_ = 0.60 (Ethylacetate: Hexane = 50:50);mp: 228–230 °C; FT-IR (cm^−1^): 3388, 3272, 2950, 1738, 1548, 1484, 1315; ^1^H NMR (500 MHz, DMSO-*d*_6_) (*δ*, ppm): 13.82 (s, 1H, N-*H*, pyrazole ring), 11.21 (s, 1H, N*H*, hydrazone), 10.02 (s, 1H, N-*H*, Isatin ring), 7.42 (d, *J* = 7.4 Hz, 1H, Ar-*H*), 7.38 (d, *J* = 2.0 Hz, 1H, Ar-*H*), 7.32 (d, *J* = 1.6 Hz, 2H, Ar-*H*), 7.15 (d, *J* = 2.1 Hz, 1H, Ar-*H*), 6.92 (d, *J* = 8.0 Hz, 2H, Ar-*H*), 6.80 (d, *J* = 2.8.0 Hz, 1H, Ar-*H*), 3.91 (s, 3H, OC*H_3_*). ^13^C NMR (125 MHz, DMSO-*d*_6_) (*δ*, ppm): 164.4, 163.6, 159.1, 132.7, 132.7, 132.5, 129.9, 127.1,126.6, 123.8, 115.5, 114.3, 57.1; ESI-MS (*m*/*z*) calcd. for C_19_H_14_BrN_5_O_3_: 439.0280 Found: 440.10 [M+H]^+^.

#### 4.1.19. (E)-3-(3-Bromophenyl)-N′-(7-Chloro-2-Oxoindolin-3-Ylidene)-1H-Pyrazole-5-Carbohydrazide (**PS13**)

Yellow solid; yield 82%; R_f_ = 0.62 (Ethylacetate: Hexane = 50:50); mp: 205–208 °C; FT-IR (cm^−1^): 3332, 3241, 3082, 1605, 1483, 1341, 1167; 6.80; ^1^H NMR (500 MHz, DMSO-*d*_6_) (*δ*, ppm): 13.90 (s, 1H, N-*H,* pyrazole ring), 11.23 (s, 1H, N*H*, hydrazone), 10.08 (s, 1H, N-*H*,_._ Isatin ring), 7.42 (d, *J* = 7.4 Hz, 2H, Ar-*H*), 7.38 (d, *J* = 2.0 Hz, 2H, Ar-*H*), 7.32 (d, *J* = 1.6 Hz, 1H, Ar-*H*), 7.15 (d, *J* = 2.1 Hz, 2H, Ar-*H*), 6.92 (d, *J* = 8.0 Hz, 2H, Ar-*H*), ^13^C NMR (125 MHz, DMSO-*d*_6_) (*δ*, ppm): 164.3, 163.5, 143.8, 140.9, 134.0, 133.9, 130.2, 127.9, 125.7, 121.5, 114.7; ESI-MS (*m*/*z*) calcd. for C_18_H_11_BrClN_5_O_2_: 442.9785 Found: 443.25 [M+H]^+^.

#### 4.1.20. (E)-3-(3-Bromophenyl)-N′-(6-Nitro-2-Oxoindolin-3-Ylidene)-1H-Pyrazole-5-Carbohydrazide (**PS14**)

Light brown; yield 75%; R_f_ = 0.72 (Ethylacetate: Hexane = 50:50);mp: 225–228 °C; FT-IR (cm^−1^): 3325, 3253, 3087, 1741, 1536, 1463, 1343, 1169; ^1^H NMR (500 MHz, DMSO-*d*_6_) (*δ*, ppm): 13.88 (s, 1H, N-*H*, pyrazole ring), 11.32 (s, 1H, N*H*, hydrazone), 10.09 (s, 1H, N-*H*, Isatin ring), 7.42 (d, *J* = 7.4 Hz, 1H, Ar-*H*), 7.38 (d, *J* = 2.0 Hz, 1H, Ar-*H*), 7.32 (d, *J* = 1.6 Hz, 3H, Ar-*H*), 7.15 (d, *J* = 2.1 Hz, 1H, Ar-*H*), 6.92 (d, *J* = 8.0 Hz, 2H, Ar-*H*); ^13^C NMR (125 MHz, DMSO-*d*_6_) (*δ*, ppm): 164.2, 163.4, 148.8, 145.9, 133.3, 131.8, 130.9, 130.4, 129.4, 127.0, 125.2, 124.3, 124.2, 121.7; ESI-MS (*m*/*z*) calcd. for C_18_H_11_BrN_6_O_4_: 454.0025 Found: 455.08 [M+H]^+^.

### 4.2. Pharmacological Evaluation

#### 4.2.1. Drug-Likeliness Assessment

The physico-chemical parameters for all the compounds (**PS1–14**) were evaluated using freely available tool SwissADME by calculating their Lipinski’s Ro5 (Rule of five) descriptors [41].

#### 4.2.2. In Vitro Inhibition Study of Methionine Aminopeptidases (MetAPs)

MetAP appears to serve a vital role in the co- or post-translational modification of proteins, making it essential for bacterial cell metabolism and survival [42]. In this study, all enzymes, which were used for this study, including *Mt*MetAP1c (*M. tuberculosis)*, *Ef*MetAP1a (*E. faecalis)*, *Sp*MetAP1a (*S. pneumoniae)* and *Hs*MetAP1b (*H. sapiens)* enzymes, were expressed and purified in our lab as previously reported [43]. All compounds prepared 10 mM stocks in DMSO and were used for enzyme inhibition studies. The concentration of each enzyme is adjusted such that they have similar reaction rates (*Hs*MetAp1b, 1 µM; *Sp*MetAP1a, 5 µM; *Ef*MetAP1a, 1 µM; *Mt*MetAP1c, 1 µM;) as reported earlier [43]. All assays were performed in 100 mL reaction containing 50 mM HEPES (pH 7.5 for *Sp*MetAP1a and *Hs*MetAP1b; 8.0 for *Ef*MetAP1a, *Mt*MetAP1c, *Hs*MetAP1b), 150 mM NaCl, and CoCl_2_ (three molar equivalents of corresponding enzymes). The reaction was started by adding 50 μM L-Methionine 7-amido-4-methyl coumarin (Met-AMC) and continuously monitoring the fluorescence of released 7-amino-4-methyl coumarin (AMC) (380 nm excitation and 460 nm emission) in a microplate multimode reader (TECAN, Austria). The maximum percentage of inhibition against enzymes is calculated at 10 mM compound concentration. Which are compounds shown the 40% or more inhibition against enzymes at the concentration of 10 µM are further determined for their *K_i_* value. The *K_i_* values were determined according to the Morrison method using three different concentrations of substrates and ten different concentrations of the inhibitor. All experiments were performed in triplicate. GraphPad Prism version 8.0.0 (San Diego, CA, USA) was used to determine all the kinetic parameters.

#### 4.2.3. Screening of Compounds

In vitro screening of the compounds (**PS1–14**) was performed to assess the antibacterial activity against three Gram-positive (*Staphylococcus*
*aureus, Enterococcus faecalis* and *Bacillus subtilis,*) and three Gram-negative (*Pseudomonas aeruginosa, Klebsiella pneumonia* and *Escherichia coli*) bacterial strains using the standard NCCLS broth dilution assay. The study’s standard drugs were ciprofloxacin (CIP) and ampicillin (AMP). Compounds were dissolved in a polar solvent (DMSO) and distributed into a 96-well plate in 200 µL of nutrient broth medium, with a final concentration of the compounds 250 g/mL in each well. The bacterial cells (about 2.5 × 10^5^ cells/mL) were inoculated into each well and incubated at 37 °C overnight. After incubation, the growth was measured by turbidity of the culture at 590 nm by spectrophotometer [44].

#### 4.2.4. Minimum Inhibitory Concentration (MIC)

The MIC of the selected compounds **PS3**, **PS6**, **PS9** and **PS11** against the above-mentioned strains was determined after the growth inhibition assay. Briefly, the culture was taken in exponential phase (OD_600_) = 0.5 using the conventional broth dilution method in a 96-well plate with a cell concentration of 2 × 10^5^ CFU/mL. The 96-well plates were kept in an incubator shaker at 100 rpm for 24 h. The MIC was determined as the lowest concentration that prevented observable cell growth. All experiments were conducted with three biological triplicates [45,46].

#### 4.2.5. Minimum Bactericidal Concentration

Minimum bactericidal concentration (MBC) was used to identify the lowest concentration at which an antimicrobial agent can kill a specific microbe, as previously explained. Aliquots (10 μL) from each well that revealed no microbe growth were plated on Mueller-Hinton Agar (MHA) and incubated at 37 °C to determine Minimum Bactericidal Concentration. The MBCs were chosen as the lowest concentration that produced no growth following sub-culturing. It was calculated as the maximum dilution at which the bacterial inoculum failed to grow in the medium. The ability of a compound to exhibit antibacterial activity is measured as the ratio of MBC to MIC. The compound was classified as bactericidal if the MBC/MIC ratio was less than 2, and bacteriostatic if the ratio was between 2 and 16 [37,47].

#### 4.2.6. Disc Diffusion Assay

This study was used to assess the antibacterial efficiency of the test compounds **PS3**, **PS6**, **PS9** and **PS11**. Discs with doses equivalent to ½ MIC, MIC, and 2MIC were placed on a solid nutrient agar medium and incubated overnight at 37 °C. After incubation, the zone of inhibition (mm) was recorded and analyzed as explained in Ali et al. [46].

#### 4.2.7. Combination Study

**PS3**, **PS6**, **PS9** and **PS11** were tested in combination with ampicillin and ciprofloxacin against bacterial strains (*K. pneumoniae, E. faecalis, S. aureus, P. aeruginosa, B. subtilis and E. coli)* to determine if they had synergistic antibacterial action using the method of Wani et al. [26]. If the FICI value is less than or equal to 0.5, it indicates that the selected compounds are synergistic, and if it is in between 0.5 and 4, it has an indifferent impact, and if it is higher than 4, it is an indication of antagonism.

#### 4.2.8. Growth Kinetics

Based on the above experiments, **PS9** showed the best activity among tested compounds. A growth kinetic assay was also performed to ascertain how **PS9** affected the growth of the tested strains (*E. coli*, *E. faecalis, K. pneumonia* and *P. aeruginosa*) in contrast to ampicillin. Bacterial strains were tested as mentioned in Habib et al. [48]. Briefly, ½ MIC, MIC and 2MIC values of **PS9** were taken to assess the growth of bacterial strains whereas untreated cells and media were taken as positive and negative control, respectively.

#### 4.2.9. Time–Kill Curve Assay

A time–kill curve study was performed against bacterial strains *(E. coli*, *E. faecalis, K. pneumonia* and *P. aeruginosa*) to determine whether compound **PS9** is bactericidal or bacteriostatic. MIC, 2MIC and 4MIC concentrations were employed for the time–kill assay as explained by Ali et al. [46]. Untreated bacterial cells were used as a control. 100 µL aliquot was taken from each flask at regular intervals. An aliquot was further diluted in phosphate buffer saline solution and from each concentration 50 µL aliquot was inoculated on a nutrient agar plate and spread. Colonies were counted after the incubation of the plate at 37 °C. For each strain, the average (mean) colony count data (log10 CFU/mL) was plotted against time.

#### 4.2.10. Bioassay on the Environmental Resistant Strains

**PS9** was shown to be the most promising antibacterial lead compound among the tested compounds which led us to carry out further investigations against environmental multidrug-resistant (MDR) isolates. The MIC and MBC of the compound **PS9** were determined using 21 MDR environmental isolates. We collected multidrug-resistant isolates from several environmental sources like lentic (lake and pond) and effluent (slaughterhouse, sewage treatment plant and hospital drainage) water samples. Their resistance pattern and the accession number are mentioned in Appendix A. MIC and MBC were taken from the microdilution method with the same preparation and methodology as explained above in this study.

### 4.3. Evaluation of In Vitro and In Vivo Toxicity

#### 4.3.1. Hemolytic Activity

The compound **PS9** was evaluated for hemolytic effect on human RBCs [49]. In brief, RBCs suspension 10% (*v*/*v*) cleaned with 1 × PBS and resuspended again in 1 × PBS (pH 7.4). Suspension was treated with **PS9** at 1.65, 3.31, 7.62, 15.12, 31.25, 62.5, 125, 250 and 500 µM concentrations at 37 °C for 2 h. The samples were centrifuged and absorbance of the supernatant was taken at 415 nm to obtain the percentage of RBCs lysis. Triton X-100 with 1% (*v*/*v*) was used as a positive control. Percentage of RBCs lysis was calculated as:% RBCs lysis = (OD_415nm_ sample − OD_415nm_ PBS)/(OD_415nm_ Triton X-100 1% − OD_415nm_ PBS)(1)

#### 4.3.2. Cytotoxicity Assay

**PS9** was tested for cytotoxicity against human embryonic kidney (HEK293) cells using a colorimetric MTT assay [26]. HEK293 cells were grown at 37 °C in a humidified environment of 5% CO_2_ in DMEM media enriched with 10% FBS and 1% penicillin, streptomycin solution. HEK293 cells (1 × 10^4^) were seeded in triplicate and incubated overnight in a CO_2_ incubator. After the treatment (5–200 μM) with **PS9**, the cells were kept for 48 h at 37 °C. Freshly prepared 5 mg/mL MTT was supplied to the samples and kept on incubation at 37 °C for 4–5 h. DMSO was added to solubilize the formazan crystals. A microplate reader (Bio-Rad, Hercules, CA, USA) was used for taking absorbance at 570 nm and all the experiments were done in triplicate.

#### 4.3.3. In Vivo Toxicological Assessment of **PS9** on *Galleria mellonella* Larvae

The objective of this experiment was to assess the in vivo toxicity of **PS9** using larvae of *G. mellonella*. The insect immunity system shows many similarities to the innate immune response of mammals therefore insects may be utilized in place of mammals for routine testing of microbial pathogens and for assessing the in vivo efficacy of antimicrobial agents. The results obtained using *Galleria larvae* show a strong correlation to those generated using mammals and larvae have the advantage of producing results within 48 h, being inexpensive to purchase and house, and being free from the legal and ethical restrictions that affect mammalian testing. Larvae can also be used to assess toxicity and there is a strong correlation between the toxicity of compounds in larvae and mammals.

##### **PS9** Solution Preparation

**PS9** stock solutions were prepared in DMSO (≤10%) and deionized water and maintained at room temperature.

##### *G. mellonella* Inoculation

The greater wax *G. mellonella* sixth instar larvae were purchased from Live foods Direct Ltd. Sheffield, UK, and kept in the dark at 15 °C to suspend pupation. Healthy larvae with no appearance of melanization and weighing 150–250 mg were selected and stored in sterile nine cm Petri dishes containing wood shavings.

Overnight grown bacterial culture was resuspended in PBS to obtain desired cell densities. Bacterial suspensions (20 µL) were injected into the hemocoel via the last left proleg using a U-100 insulin syringe and incubated at 37 °C. Larvae were administered 20 µL of **PS9** solutions through the last right proleg 30 min post-infection. Controls consisted of sterile water and DMSO (≤10%). Larvae were incubated at 37 °C for all studies. Survival of inoculated larvae, (*n* = 10 per group), was monitored every 24 h for up to five days. Mortality was based on melanization and loss of touch response.

##### Assessment of *G. mellonella* Hemocyte Density

Larvae were inoculated with **PS9** or water and DMSO and incubated at 37 °C. Larvae were punctured with a needle (26G × 1/2″) to extract 20 μL hemolymph per larva. Hemolymph was pooled (*n* = 3 larva per group, per time point) and diluted in 100 μL pre-chilled PBS supplemented with *N*-phenylthiourea to prevent melanization. Samples were maintained on ice before hemocyte quantification using a hemocytometer. Cell densities were determined per ml of PBS at respective time points.

## 5. Conclusions

In conclusion, a variety of novel Isatin-pyrazole hydrazone conjugates with appropriate substitutions were synthesized and tested as effective antibacterial agents against Gram-positive and negative bacterial strains. Among the synthesized compounds (**PS1-PS14**), compound **PS9** was found potential antibacterial agent towards selective bacterial strains targeting bacterial MetAPs. Interestingly, **PS9** inhibited the human analogue *Hs*MetAP1b with *K*i (631.7 µM) about ten thousand-fold higher than the bacterial MetAPs confirming its efficacy and selectivity. Additionally, **PS9** displays synergistic impact when combined with AMP or CIP against aforementioned strains. Furthermore, cytotoxicity testing demonstrated that the compound **PS9** was non-cytotoxic in the HEK293 cell line at concentrations up to 200 µg/mL. **PS9** caused a small increase in hemocyte density, indicating a stress reaction, and was determined to be safe for *G. mellonella* larvae in vivo at concentrations up to 1000 mg/mL. Moreover, the compound **PS9** showed inhibitory action against environmental multidrug-resistant strains and even better efficacy against the selective bacterial isolates as compare to the standard drug used. Overall, the biochemical and various cell-based assays performed in this study demonstrate that **PS9** compound has the potential to be developed as an effective and safer antibacterial agent.

## Data Availability

Data is contained within the article.

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
