# Peer review of "Design, Synthesis and Mechanistic Studies of Novel Isatin-Pyrazole Hydrazone Conjugates as Selective and Potent Bacterial MetAP Inhibitors"

_antibiotics, 2022, doi:10.3390/antibiotics11081126_

Round 1

Reviewer 1 Report

The authors describe using Isatin-pyrazole hydrazone conjugates with a series of substitutions and tested them as effective antibacterial agents against bacterial strains (Gram-positive and negative). PS9 was found as the potential antibacterial agent for sensitive strains of Gram-positive and Gram-negative bacteria, it showed non-cytotoxic in human cells up to 200 μg/mL. Also, the biochemical analysis showed PS9 has potential inhibition activity against prokaryotic MetAP enzyme but not for human MetAP. Overall, the manuscript is very well written, the medchem was properly conducted and a very interesting approach is also presented. Compounds are well characterized both from the chemical and the biological point of view. So, I am supportive of the manuscript for publication in Antibiotics.

However, there are several questions that need to be revised:

1.     The structure of PS9 in Figure 1 and the structure of PS1-14 in Scheme 1 were drawn not appropriate, the carbon bond of hydrazone should link with N instead of H, please correct it.

2.     From the growth inhibition profile Table2, PS11 showed strong inhibition, but it showed lower activity in the biochemical Ki assay, please give some explanation on this disconnection.

Author Response

General Comment:

We appreciate the critical reading and deep indulgence of the esteemed reviewer. We found these comments highly helpful to remove errors and make the presentation of this work scientifically sound and flawless.   

Specific comments:

  1. As suggested to improve the English language, we have revised the manuscript extensively and tried to remove any grammatical or typos throughout the manuscript. Al the changes are highlighted in the track changes mode. 
  2.  The chemical structure of PS9 has been corrected and revised Figure 1 and Scheme 1 have been provided in the revised manuscript.  
  3. As we have mentioned in the manuscript, we initially screened all the compounds at a single concentration of 250 µg/mL to check their activity and sort them out. We have taken a relatively high concentration (250 µg/mL, which may incur PS11 as a strong inhibitor. In contrast, in the biochemical assay, the concentration of the compound PS11 is only 10µM, which might be responsible for the low activity of the compound. The possibility of another molecular target can not also be ruled out.  We are working on optimization of this class of compound with varied substitution at another side of the scaffold to improve its efficacy, selectivity, and toxicity issue if any.   

Reviewer 2 Report

Irfan et al in article Design, synthesis and mechanistic studies of novel isatin-pyrasole hydrazone conjugates as selective and potent bacterial MetAP inhibitors puts an accent on PS9 of other 13 tested compounds. If it has an effect on bacteria which are not affected by standard atibiotics, this could represent an important edition to the existing therapies. The authors performed a range of experiments ( MIC, MBC, disc diffusion assay, growth curve, time-kill curve) in order to confirm the growth inhibitory properties of these compounds. The article is well written with very minor English/spelling errors (page 3 line 90; page 9 line 231; page 15 line 368).

Irfan and colleagues are describing the design and potential usage of isatin-pyrozole hydrazone conjugates as a novel potent bacterial MetAP inhibitors. There ia an importance of the topic they address, given the worldwide problem with antibiotic resistance. Among 14 compounds described (PS1-14), PS9 seems to distinguish itself as the candidate with highest level of action against multidrug-resistant Gram-positive and Gram-negative bacteria; but also other of them were suggested to inhibit MetAPs from bacteria such as Enterococcus faecalis, Streptococcus pneumonie, Mycobacterium tuberculosis. The article brings novel insight into synergetic action of these MetAP inhibitors with standard antibiotics (PS6 and PS9 combined with ampicillin and ciprofloxacin) against several bacterial strains. Additionally, authors supported their claims by performing MBC, disc diffusion assay, growth curve and time-kill curve experiments. Tables and figures are very clear and appropriately used, and the methodology is described in many details.  The aspect that could be addressed in the author`s future articles would be how to optimize PS9 for the human usage, given the fact that Ki values rather differ from those used on procaryotic cells. Minor: Few English/misspellings were noticed.

Author Response

General Comment:

We are highly thankful to the learned reviewer for their critical comments and for appreciating the work. Very nicely, the work has been summarized by the reviewer showing the deep involvement and interest in the work. We have taken each and every comment of the learned reviewer very seriously to make the work more presentable, scientifically sound, and flawless.

Specific comments:

  1. As pointed out, very minor English/spelling errors (page 3 line 90; page 9 line 231; page 15 line 368) have been corrected in the revised manuscript. 
  2. We totally agree with the learned reviewer about the significance of the work. Further, we are working on optimization of this class of compound especially PS9 with varied substitutions at another side of the scaffold to improve its efficacy, selectivity, and toxicity issue if any. We are also trying hard to do collaborative in vivo studies in mouse models and other pre-clinical studies.